# Evaluation of a Balloon Implant for Simultaneous Magnetic Nanoparticle Hyperthermia and High-Dose-Rate Brachytherapy of Brain Tumor Resection Cavities

**DOI:** 10.3390/cancers15235683

**Published:** 2023-12-01

**Authors:** Shuying Wan, Dario B. Rodrigues, Janet Kwiatkowski, Omaditya Khanna, Kevin D. Judy, Robert C. Goldstein, Marty Overbeek Bloem, Yan Yu, Sophia E. Rooks, Wenyin Shi, Mark D. Hurwitz, Paul R. Stauffer

**Affiliations:** 1Department of Radiation Oncology, Thomas Jefferson University Hospital, Philadelphia, PA 19107, USA; yan.yu@jefferson.edu (Y.Y.); sophie.e.rooks@gmail.com (S.E.R.); wenyin.shi@jefferson.edu (W.S.); paulstauffer@gmail.com (P.R.S.); 2Department of Radiation Oncology, University of Maryland School of Medicine, Baltimore, MD 21201, USA; drodrigues@som.umaryland.edu; 3MAE Group, Deerfield, NH 03037, USA; janetk@maegroups.com; 4Department of Neurosurgery, Thomas Jefferson University Hospital, Philadelphia, PA 19107, USA; goldey.khanna@gmail.com (O.K.); kevin.judy@jefferson.edu (K.D.J.); 5AMF Life Systems, Auburn Hills, MI 48326, USA; rcgoldstein@amflife.com; 6Phoenix DeVentures, Morgan Hill, CA 95037, USA; martyb@phoenixdeventures.com; 7Radiation Medicine, Westchester Medical Center University Hospital, Valhalla, NY 10595, USA; mark.hurwitz@wmchealth.org

**Keywords:** thermobrachytherapy balloon, magnetic nanoparticle hyperthermia, nanofluid, high-dose-rate brachytherapy, human skull model, gel brain phantom, pig brain, *in vivo* thermometry, glioblastoma

## Abstract

**Simple Summary:**

Glioblastoma (GBM) generally recurs locally with a dismal median survival of <18 months. Combining thermal therapy with radiation therapy enhances radiation response and improves clinical outcomes. This study evaluates a thermobrachytherapy balloon implant intended for the simultaneous heat and radiation treatment of tumor resection cavities. The data demonstrate that our prototype implant produces spherically symmetric heat and radiation dose distributions around the balloon, while the *in vivo* experiments confirm our ability to heat ≥40 °C at a 5 mm distance from the balloon surface in highly perfused pig brain tissue. The device is now ready for the finalization of regulatory approvals in anticipation of early-stage clinical investigation.

**Abstract:**

Previous work has reported the design of a novel thermobrachytherapy (TBT) balloon implant to deliver magnetic nanoparticle (MNP) hyperthermia and high-dose-rate (HDR) brachytherapy *simultaneously* after brain tumor resection, thereby *maximizing* their synergistic effect. This paper presents an evaluation of the robustness of the balloon device, compatibility of its heat and radiation delivery components, as well as thermal and radiation dosimetry of the TBT balloon. TBT balloon devices with 1 and 3 cm diameter were evaluated when placed in an external magnetic field with a maximal strength of 8.1 kA/m at 133 kHz. The MNP solution (nanofluid) in the balloon absorbs energy, thereby generating heat, while an HDR source travels to the center of the balloon via a catheter to deliver the radiation dose. A 3D-printed human skull model was filled with brain-tissue-equivalent gel for in-phantom heating and radiation measurements around four 3 cm balloons. For the *in vivo* experiments, a 1 cm diameter balloon was surgically implanted in the brains of three living pigs (40–50 kg). The durability and robustness of TBT balloon implants, as well as the compatibility of their heat and radiation delivery components, were demonstrated in laboratory studies. The presence of the nanofluid, magnetic field, and heating up to 77 °C did not affect the radiation dose significantly. Thermal mapping and 2D infrared images demonstrated spherically symmetric heating in phantom as well as in brain tissue. *In vivo* pig experiments showed the ability to heat well-perfused brain tissue to hyperthermic levels (≥40 °C) at a 5 mm distance from the 60 °C balloon surface.

## 1. Introduction

Glioblastoma (GBM) generally recurs locally [1,2,3] with a dismal median survival of <18 months [4]. The current standard of care for newly diagnosed GBM involves the upfront maximal resection of tumors, or a simple biopsy for non-resectable tumors, followed by chemoradiotherapy and maintenance temozolomide with Tumor-Treating Fields [5].

Combining thermal therapy (40–48 °C for 15–60 min) with radiation therapy enhances radiation response [6,7,8,9,10] and improves clinical outcomes [11,12,13,14,15]. One randomized clinical trial in primary GBM demonstrated a statistically significant doubling of 2-year survival (31% vs. 15%, *p* = 0.02) by adding two 30 min interstitial microwave hyperthermia sessions to an interstitial iodine-125 (^125^I) brachytherapy boost, following the then standard partial brain external beam radiotherapy (EBRT) with 59.4 Gy in 33 fractions and oral hydroxyurea as a radiosensitizer [16]. This excellent result published in 1998 was among the best clinical outcomes for GBM until a recent trial that added TTFields to the current standard of care [17]. Unlike TTFields, which is now a commercial product, the interstitial microwave hyperthermia approach was never optimized or translated into clinical practice. 

Other clinical trials have demonstrated a positive benefit of adjuvant hyperthermia for brain tumors [11,12,13,14,18,19,20,21]. Treatment approaches that have been investigated involve interstitial heating technologies that can focus heat locally within an at-risk tissue volume at depth in brain. Techniques used to generate moderate temperature hyperthermia include miniature implantable radiofrequency electrodes [22,23,24], microwave antennas [25,26,27,28], hot source techniques such as DC voltage heated wires [29,30], and magnetic field excited ferromagnetic seeds [31,32] or needle-injected magnetic nanoparticles (MNPs) [33,34,35,36]. The design and typical performance of the wide range of equipment used for heating brain tumors have been reviewed previously [37,38,39]. With the exception of the injected magnetic nanoparticles approach [33,34] using commercial equipment approved in 2010 by the European Medicines Agency, past hyperthermia trials relied on prototype heating equipment that was approved for use at only one research institution. Despite promising results, none of the other devices were commercialized for widespread use.

While each of these technologies has demonstrated the ability to heat local regions of brain around implanted catheters or needle-injected MNPs, each of the techniques has technical challenges in delivering uniform heat to entire at-risk tissue volumes and combining the heat with optimal timing and spatial conformity with the radiation dose. Although laboratory data provide a strong rationale for combining heat simultaneously with radiation for maximum synergism [6,40], the simultaneous delivery of heat and radiation has not been possible using existing devices and treatment protocols. Instead, previous hyperthermia approaches have involved inserting heat sources (RF electrodes, microwave antennas, resistance wires, ferromagnetic seeds, or nanoparticles) within an irregularly spaced array of percutaneously placed catheters or needle insertion tracts. The heat sources must then be removed in order to place interstitial radioactive seeds within the same 1–2 cm spaced array of catheters. Thus, the heat and radiation doses have always been separated in time, reducing the potential thermal enhancement. The larger problem with all these interstitial approaches is in delivering uniform doses of heat and radiation to an approximately spherical or annular tumor target from an irregularly spaced array of non-parallel catheters that are usually spaced too far apart. Moreover, the catheters must be inserted through small burr holes in the skull while carefully avoiding critical blood vessels and brain structures. These constraints on implant array geometry have always led to unavoidable cold and hot regions of the heat and radiation dose distributions. These interspersed regions of under- and over-dosed tissue reduce treatment effectiveness and potentially increase complication rates.

After decades of investigation of fundamentally imperfect percutaneous catheter-based treatments not well matched to the task of treating a large annular tumor margin surrounding a resection cavity at depth in sensitive normal brain, the development of a novel treatment approach was initiated. Stauffer et al. [41] describe the design and early testing of a novel dual-wall balloon implant intended for the simultaneous delivery of heat and radiation doses applied uniformly to the resection cavity wall and falling off within 5–10 mm of the tumor margin in order to minimize complications in surrounding brain. The current effort presents a thorough evaluation of two different size and style thermobrachytherapy (TBT) balloon implant devices manufactured according to that design. These devices are evaluated for the robust and stable performance of compatible heat and radiation delivery components. Thermal dosimetry performance is characterized for 3 cm and 1 cm diameter TBT balloons implanted in a human-sized brain phantom and *in vivo* pig brains, respectively. Radiation doses around 3 cm diameter balloons are compared with and without (i) MNP inside the balloon, (ii) magnetic field around the balloon, and (iii) high temperatures in and around the balloon as expected during simultaneous heat and radiation dose delivery. This preclinical evaluation of the TBT balloon device is performed as a critical step toward the final approval and initiation of Phase I clinical trials of the simultaneous heat and brachytherapy treatment of resectable brain tumors. 

## 2. Materials and Methods

The components of the complete TBT system are described in the following section. After that, methods are given to evaluate the TBT balloon in laboratory phantom studies regarding its durability, compatibility, and thermal and radiation dosimetry. The final section is focused on *in vivo* pig brain thermal dosimetry studies for preclinical validation of the feasibility of the TBT heating of highly perfused brain tissue. 

### 2.1. Thermobrachytherapy System Components

The dual-modality TBT therapy system has five components: (1) an expandable spherical balloon implant to be filled with MNP solution (nanofluid), (2) a radiofrequency (RF) generator, (3) a head-sized RF induction coil to generate the magnetic field that activates heating within the nanofluid, (4) a fiber optic probe to monitor the nanofluid temperature inside the balloon, and (5) a high-dose-rate (HDR) brachytherapy delivery system. 

#### 2.1.1. TBT Balloon Implant

This study investigates two TBT balloon implant designs: a 3 cm diameter dual-wall balloon for human-sized brain phantom experiments (Figure 1A) and a 1 cm diameter single-wall balloon to fit into the much smaller pig brain (Figure 1B). 

The dual-wall balloon comprises an inner balloon filled with saline to 2 cm diameter (nominal volume 4.2 mL) and a 3 cm diameter outer balloon that forms a 5 mm thick annular layer around the inner balloon. The 5 mm out layer is filled with nanofluid (nominal volume 9.9 mL). The dual-wall design aims to reduce the nanofluid cost for larger-size treatment implants with diameters ≥3 cm. As shown in Figure 1A, multiple catheters are housed within a flexible sheath, which extends from inside the balloon to ports outside the skull: a central lumen allows the insertion of a catheter through which an HDR brachytherapy source travels to the balloon center; a second lumen allows the insertion of a fiber optic probe into the outer balloon to monitor the nanofluid temperature during heating; and two additional lumens are used to fill the inner and outer balloons. Further details of the TBT implant design are reported in reference [41].

The smaller 1 cm single-wall balloon implant is shown in Figure 1B with a nominal volume of 0.5 mL. Brachytherapy was not delivered or studied in the pig brain experiments because (1) pigs are not allowed in our clinical HDR brachytherapy suite per state and hospital regulations, and (2) brachytherapy dosimetry around a single radioactive source is well characterized regardless of water or tissue loading. Consequently, the 1 cm balloon design was simplified to include only two ports: one for inserting a fiber optic probe and the other to fill the balloon with nanofluid.

#### 2.1.2. Nanofluid and Magnetic Field Generator

In this study, we used two magnetic nanoparticle Fe_3_O_4_ solutions manufactured by Ferrotec (Bedford, NH, USA). Formulation EMG 308 (1.2% *v*/*v*; density of 1050 kg/m^3^) was used to fill the 5 mm outer layer of the 3 cm balloon, while formulation EMG 304 (4.5% *v*/*v*; density of 1220 kg/m^3^) was used in the 1 cm balloon. The latter has a significantly higher MNP concentration to compensate for the reduced heating capacity of a 1 cm balloon compared to the nanofluid volume in a 3 cm balloon (0.5 vs. 9.9 mL). The EMG 304 and 308 nanoparticles have a spherical shape with an average size of 10 nm. Key technical data of both formulations are listed in Table 1; more information can be accessed in the technical datasheets available at https://ferrofluid.ferrotec.com (accessed on 7 November 2023).

The magnetic induction system used in this study was designed by AMF Life Systems (Auburn Hills, MI, USA) [42] and is shown in Figure 2. This system is housed in a wooden table measuring 76 × 152 × 91 cm (width × length × height) with 10 cm high plate casters to make the system mobile. A head-sized 133 kHz RF induction coil is mounted vertically on top of the table surface, and a programmable RF power generator is located on a shelf underneath, together with a circulating water-cooling system (5.5 L/min maximum). The head coil has an elliptically shaped opening of 26 × 20 × 31 cm (width × length × height). The maximum power of the generator is 7.5 kW.

The RF power sequences for the experiments in this study were programmed empirically and are summarized in Table 2. The magnetic field strength (H) was controlled by changing the relative power P%, i.e., percentage of maximum amplifier power. As a safety measure, P% was limited to 63%, which generated a magnetic field strength of H = 8.1 kA/m at the center of the RF coil at our operating frequency of 133 kHz. As H is proportional to the square root of P, H can be calculated using H = sqrt(P%/63%) × 8.1 kA/m, e.g., P% = 47% corresponds to H = 7.0 kA/m.

#### 2.1.3. Fiber Optic Thermometry System

The fiber optic system used to measure the internal balloon temperature was manufactured by Photon Control (Richmond, BC, Canada) and distributed by Micronor LLC (Camarillo, CA, USA). The sensor and controller model numbers are FTP-SA3-ST1-03M and FTC-DIN-GT-HT-ST-0673, respectively. With a temperature range of 0 to 80 °C, the probe has a 1.0 mm diameter and 3 m length. The user interface was developed by Phoenix DeVentures (Morgan Hill, CA, USA) to display temperature vs. time, as well as cumulative thermal dose expressed as Cumulative Equivalent Minutes at 43 °C (CEM43) [43]. 

#### 2.1.4. HDR Brachytherapy System

An Elekta microSelectron V3 system was used to deliver HDR brachytherapy radiation in all laboratory radiation dosimetry experiments. It contains a single source welded to the end of a 2022 mm long drive cable made of stainless steel. The source is solid Iridium-192 (^192^Ir) in a capsule measuring 0.9 mm in diameter by 4.5 mm in length (about the size of a rice grain). A new source usually comes with an activity of ~10 Ci (370 GBq) and is replaced every 2.5–3 months to match its half-life of 73.8 days. The microSelectron HDR system has 18 channels; each can be connected via a transfer tube to a catheter placed within a disease site. A computerized treatment planning system is generally used to generate a plan to deliver a prescribed radiation dose to the tumor while sparing normal surrounding tissue. During treatment, the drive cable sends the HDR source precisely to each programmed position (dwell position) of each channel for a predetermined treatment time (dwell time). In this work, the HDR system was connected to a TBT balloon via one standard transfer tube, which was attached via a twistlock connector to channel 1 of the Elekta system on one end and to a 6F catheter within the central lumen of the TBT balloon on the other end. After discussion with experts at Elekta and an intensive search in the market, a commercial 6F catheter, 110230 ProGuide 6F × 294 mm, was identified as being the most compatible, robust, and cost-effective. 

### 2.2. Laboratory Studies of TBT Balloon Implant in Phantom

Four 3 cm TBT balloon implants were evaluated in the laboratory for thermal dosimetry and radiation dosimetry studies. Dimensions of the balloons were measured with a caliper in terms of the axial diameter (D_A_) and longitudinal diameter along the shaft (D_L_). D_L_ tends to be a few mm (~10%) longer than D_A_. 

#### 2.2.1. Human Skull Model and Brain-Tissue-Equivalent Gel Phantom

A gel was prepared to mimic brain tissue using 93% deionized H_2_O (DI water), 6.545% TX-151 powder (Oil Center Research, Lafayette, LA, USA), and 0.455% NaCl (*w*/*w*). This gel formulation results in an electrical conductivity of 0.80 S/m at 1.9 MHz and thermal conductivity of 0.56 W/m/K, which are similar to brain tissue [44]. A virtual skull model was retrieved from a previous study [45] and modified in COMSOL Multiphysics software (COMSOL, Inc., Burlington, MA, USA) to contain a 3 cm balloon and be stable while resting on a table during experiments (Figure 3). This design was then 3D-printed using polylactic acid filament. The skull was split into two halves to facilitate rapid disassembly for thermal imaging of the central cross-section plane as described in Section 2.2.2. 

The procedure to load the balloon during in-phantom measurements was as follows: (1) for each experiment, we prepared a new gel phantom and poured it into the skull bottom half while semi-liquid, (2) the 3 cm TBT balloon, together with thermometry setup (described in Section 2.2.2), was secured on top of the supporting pole, (3) the skull top half was placed over the bottom half and held in place via 3 alignment pins, (4) the junction line was sealed by masking tape, (5) the gel was carefully poured into the assembled skull through a 5 cm diameter opening on the top while trying to minimize displacement of the balloon and adjacent thermal monitoring devices. We fabricated the gel right before each assembly because the gel is liquid for about 1–2 min, which allows us to fill gaps around the balloon and thermometry setup to minimize air heterogeneities in the phantom. 

#### 2.2.2. Thermal Dosimetry Measurements

The heating ability of the TBT balloons was tested in the skull phantom using the gel phantom as a tissue-equivalent load. The balloon’s internal temperature was measured using the fiber optic sensor immersed in the nanofluid, with the tip of the probe gently pressing against the balloon wall. The temperature of the surrounding gel was measured by 2 stationary probes placed at about 5 and 10 mm distances from the balloon surface, as well as 2 probes moving through 15-gauge catheters placed tangent to the balloon for thermal mapping (Figure 4A). The 1.8 mm diameter 15-gauge catheters were secured to the balloon using tape to guarantee their position relative to the balloon. Each sensor traveled from −16 to +16 mm from the balloon tangent point with a step size of 2 mm (i.e., radial distance to the balloon surface ranged from r = 0.9 to 7.6 mm). The sensor stayed at each position for about 5 s; therefore, each cycle took about 3 min. The thermal mapping probe motion was robotically controlled using a motor and in-house LabVIEW-based software. The thermal mapper and connecting catheter hubs are shown in Figure 4B. 

The assembled phantom was positioned at the center of the RF coil to surround the nanofluid with a uniform magnetic field. Field homogeneity within the coil has been reported previously [41]. The pre-programmed power sequence Job #1 (Table 2) was used to reach a peak temperature of 34 °C inside the nanofluid, starting from room temperature at 20 °C, i.e., a temperature increase of ΔT = 14 °C. At the end of each heating experiment, the phantom was removed from the coil and the two halves were separated to expose the bottom phantom surface. High-resolution 2D thermal images were taken with an infrared camera (FLIR E95, Teledyne FLIR LLC, Wilsonville, OR, USA) within 30 s after power was turned off to minimize thermal dissipation of the internal temperature distribution.

#### 2.2.3. Radiation Dosimetry Measurements

To deliver a spherically symmetric radiation dose to a 5–10 mm annular rim around the balloon, the simplest approach is to position the HDR source in a single dwell position at the center of the balloon. With this simple geometry, the dwell time can be calculated by using a classical dose calculation formulism for an isotropic point source [46]:Dwell time (s) = D × d^2^/(S_k_ × 1.11) × 3600 s/h
where D is the prescribed radiation dose (cGy), d is the distance (cm) from the HDR source to the radiation dose prescription point (5–10 mm from balloon/tissue interface), S_k_ is the air kerma strength (cGy cm^2^/h) of the HDR source, and the correction factor 1.11 is the ratio of the mass-energy absorption coefficients in water to that in air averaged over the photon energy spectrum. A new 10 Ci ^192^Ir source has an air kerma strength of S_k_ = 4.08 × 10^4^ cGy cm^2^/h. As examples, to deliver 700 cGy at 5 and 10 mm distances from a 3 cm balloon surface would require 223 and 348 s dwell times, respectively, while 200 cGy at 5 and 10 mm distances from the balloon surface would require 64 and 99 s dwell times, respectively. In this study, we arbitrarily chose to investigate a dose of 500 cGy at the balloon surface, producing ~200 cGy at a 10 mm distance.

The effect of having nanofluid in the balloon around the HDR source on the radiation dose delivered to tissue was investigated in three scenarios using the 3 cm diameter dual-wall balloon with the inner chamber filled with DI water. In the first case, the radiation dose was measured and compared with the outer chamber filled with either DI water or nanofluid, both surrounded by air. In the second and third experiments, the radiation dose was measured with the outer chamber filled with nanofluid, comparing HDR alone versus HDR + HT, where the RF magnetic field was turned on to activate nanofluid heating during the radiation dose measurements. In experiment 2, the balloons were surrounded by air and in experiment 3, the balloons were immersed in the gel brain phantom, as shown in Figure 4. Experiment 1 evaluates the effect of the surrounding nanofluid on the radiation dose, while experiments 2 and 3 evaluate the effects of the magnetic field and heating of the balloon and/or surrounding gel phantom on the radiation dose. 

The radiation dose was measured as the absolute point dose by taping optically stimulated luminescent dosimeters (OSLDs) at the outer surface of the balloon, while 2D dose distributions were measured by placing a Gafchromic EBT3 dosimetry film underneath the balloon lying on a table. The OSLDs used were nanoDot dosimeters (LANDAUER, Inc., Glenwood, IL, USA) that consisted of thin chips with an area of about 1 × 1 cm^2^. Both OSLD chips and Gafchromic film are for single use only.

In order to measure the radiation dose 10 mm from the balloon surface, mimicking the dose to the at-risk tumor margin 10 mm distant from the tumor resection cavity wall, a piece of 1 cm thick superflab bolus was sandwiched between balloon #3 and an OSLD. These measurements were carried out with HDR alone and then with HDR + HT. They were also compared with readings from an OSLD positioned directly underneath the balloon.

Different RF power sequences were programmed for in-air and in-phantom measurements (Table 2). For in-air measurements, we started with Job #2 to produce a maximum temperature T_max_ of up to 77 °C inside the nanofluid for a test period of 12 min. On the following day, the test was repeated with reduced power (Job #3) and T_max_ decreased to 55 °C, which was closer to what would be used in a hyperthermia treatment. For in-phantom measurements, due to rapid thermal dissipation from the balloon to the surrounding gel, we used a higher field strength, as described in Job #4, to reach T_max_ ~50 °C.

### 2.3. In Vivo Experiments of TBT Balloon Implant in Pig Brain

All animal studies were performed in an animal surgery room under a protocol approved by the Institutional Animal Care and Use Committee (IACUC) at Thomas Jefferson University. Three live female pigs ranging from 40–50 kg were supplied by a local vendor (Animal Biotech Industries, Inc., Doylestown, PA, USA). Experiments were performed in a time span of 4 weeks. 

Once the animal was anesthetized, the neurosurgeon removed a section of the scalp and drilled through the thick skull to expose the brain, as shown in Figure 5. A portion of normal brain was resected in one hemisphere for placement of the 1 cm single-wall balloon implant. Two thermal mapping catheters were positioned tangent to the balloon, as close to parallel and 1 cm apart as possible, and all three were inserted together into the resection cavity. Finally, two stationary probes were placed at about 0 and 5–10 mm distances from the balloon surface. 

To ensure a snug fit, the balloon was deflated before insertion via a 10 mL syringe attached and inflated again once inside the resection cavity. Each thermal mapping sensor traveled in 1 mm steps from −8 to +8 mm from the balloon tangent point. Taking into account the 1.8 mm diameter of the thermal mapping catheters, the radial distance from the 1 cm diameter balloon surface to the thermal mapping sensor ranged from r = 0.9 to 4.9 mm. In order to reach and maintain a peak balloon temperature of ~60 °C during the pig brain experiments, we used RF power sequence Job #5 (Table 2). 

Immediately after each *in vivo* thermal dosimetry procedure, the animal was euthanized without ever waking from anesthesia, to ensure it was never in pain or systemic stress.

## 3. Results

### 3.1. Thermal Dosimetry in Phantom

Four 3 cm TBT balloons were filled with 5.6 mL distilled water in the inner chamber and 8.5 mL nanofluid in the outer chamber for thermometry experiments in laboratory studies. When filled, the balloons were quasi-spherical, presenting an axial diameter (D_A_) of 30 ± 1 mm and a longitudinal diameter along the shaft (D_L_) of 32 ± 1 mm. 

The balloons were first tested in air inside the RF coil under high stress conditions for about 20 min multiple times. Figure 6 shows the infrared images of one of these stress tests, noting that the nanofluid temperature recorded via the fiber optic sensor reached 60 °C. The infrared images were taken within 30 s after power off, which justifies the slightly lower maximal temperature recorded (57.7–59.3 °C). All balloons endured the stress tests without any apparent degradation from the heat. The balloon wall material was also unaltered from visual inspection and tactile perception.

A time–temperature profile is shown in Figure 7 for a representative balloon experiment in a non-perfused brain phantom. As expected, the two thermal maps in the phantom (red and blue lines) closely follow the driving temperature measured in the nanofluid (dark blue line). The maximum thermal map temperatures are measured when the probes are closest (~1 mm) to the balloon surface and the temperatures drop quickly as the probes move away from the balloon surface, in cycles of 3 min. The temperatures of the two stationary probes also follow the nanofluid driving temperature, but with a delay due to the time required for the thermal conduction wave to propagate deeper into the phantom. The lower temperatures obtained in these distant probes are expected due to the greater radial distances from the balloon surface. The shaded yellow dashed line shows the programmed relative RF power sequence Job #1 (Table 2) that induced the nanofluid heating.

Figure 8 shows an infrared thermal image of the bottom half of the split-phantom 2 min after power off, where we can observe spherically symmetric heating that is well localized around the balloon. No heating is observed at the skull surface, which conceivably could occur from eddy currents if excessive magnetic fields were used. This lack of surface tissue heating illustrates the safety of this device since heating is restricted to the intended at-risk tumor margin near the nanofluid-filled balloon. 

### 3.2. Compatibility of MNP Hyperthermia and HDR Brachytherapy

#### 3.2.1. Effect of Nanofluid on Radiation Dosimetry

The radiation doses measured around a 3 cm diameter balloon (#2) are given in Table 3 for the experiments with both chambers filled with DI water and subsequently when water in the outer chamber was replaced with a 5 mm thick layer of nanofluid (DI water + nanofluid). Measurement uncertainty was most likely due to the limited precision of the dosimeter position when manually taping the OSLD chips on the curved non-rigid surface of the balloon. A small displacement (e.g., 0.5 mm) will lead to a significant measurement difference (6%) due to inverse square law. Nonetheless, as shown in Table 3, the radiation doses with and without nanofluid were similar to each other with an average difference of less than 10%, which would not be clinically significant.

Figure 9 shows the 2D dose distributions measured with Gafchromic film comparing two different balloon fillings: with and without nanofluid. Radiation dose profiles measured in a plane tangent to the balloon agreed with each other within 10%.

#### 3.2.2. Effect of Magnetic Field and Balloon Heating on Radiation Dosimetry

The radiation dose was measured in air using OSLDs that were placed at five locations on the surface of balloon #2 and only at one location (underneath the balloon) on the other balloons. As shown in Table 4, the radiation doses on the balloon surface were not significantly affected by the magnetic field or high balloon temperatures up to 77 °C; the differences were attributed to unavoidable small positioning uncertainties of the OSLD chips across multi-trial experiments. Table 5 shows the OSLD readings when a piece of 1 cm bolus was sandwiched between the balloon surface and the OSLD. The radiation dose at a 10 mm distance to the balloon surface agreed within 10% between HDR alone and HDR + HT.

To ensure there is no difference in radiation dosimetry with the balloon implanted in a tissue-like load rather than air, radiation dosimetry measurements were repeated with balloons buried in the gel brain phantom in the human skull model. Measurements were performed using four 3 cm diameter balloons encased in OSLDs and buried in gel. The differences in the OSLD readings between HDR and HDR + HT were less than 10%, as seen in Table 6.

### 3.3. Thermal Dosimetry in Pig Brain

Figure 10 presents the temperatures recorded along the thermal maps adjacent to the balloon and at stationary locations around the 1 cm balloon in three live pig brains. Within 12 min of heating, the maximum temperatures measured by the thermal mapping probes ~1 mm from the balloon surface (TM1 and TM2) ranged from 42 to 50 °C for the three pigs, i.e., the temperature increase ΔT above the core temperature (36 °C) ranged from 6 to 14 °C. 

In Pig 3, the minimum thermal mapping temperatures (TM1 and TM2) at either end of the maps (i.e., r~5 mm from balloon surface) were about 40 °C, with one end at about 41–44 °C and the other end at about 38–40 °C. This asymmetry was probably due to an asymmetrical travel range that deviated by ~1 mm from the planned travel range of −8 to +8 mm from the balloon tangent point, resulting in one end at r~4 mm (41–44 °C, i.e., average ~42.5 °C) and the other end at r~6 mm (38–40 °C, i.e., average ~39 °C) from the balloon surface. Based on this, we can reasonably conclude that the average temperature at r~5 mm from the balloon surface reached ≥40 °C.

## 4. Discussion

Our proposed dual-modality local RT + HT therapy requires the placement of a balloon in the cavity at the time of tumor resection followed by several fractions of HDR brachytherapy with simultaneous magnetic field heating. This approach focuses treatment on microscopic tumor cells immediately next to the resection margin. The thermal enhancement ratio due to elevated temperature can be as high as 5.0 [40], making tumor cells in the at-risk tumor margin more susceptible to radiotherapy. Therefore, the total dose of radiation administered can be reduced with the same effect on tumor cells around the balloon implant. This dose reduction should, in turn, reduce toxicities in surrounding normal brain such as radiation necrosis and late cognitive decline.

Two different TBT balloon implant designs were evaluated: 3 cm diameter dual-wall balloons for human-sized brain phantom experiments and smaller 1 cm diameter single-wall balloons to fit into pig brain for *in vivo* experiments (Figure 1). 

The 3 cm balloons went through laboratory studies repeatedly at high temperatures above our intended clinical use and all implants maintained their mechanical stability. The heating experiments (Figure 7 and Figure 8) using these balloons in a non-perfused gel brain phantom showed spherically symmetric heating around the balloon and no heating at the skull surface, illustrating the safety of the device as heating was restricted to the intended margin adjacent to the balloon, which mimics the at-risk tumor target region. The radiation experiments using these balloons in air (Figure 9, Table 3, Table 4 and Table 5) and in the gel brain phantom (Table 6) demonstrated the compatibility of the thermal and radiation components of the balloon implants as well as the capability of delivering both modalities simultaneously with good reproducibility and durability. The radiation dose was not significantly affected by the presence of a 5 mm layer of nanofluid, a 133 kHz magnetic field with a maximal magnetic field strength of 8.1 kA/m (typical range 3–7 kA/m), and temperatures up to 77 °C. While activation of nanoparticles with a magnetic field strength of 3–7 kA/m at 133 kHz worked well in our current approach, other researchers have used alternative techniques that may be considered in the future [47,48].

The *in vivo* pig brain experiments added multiple challenges to the thermal dosimetry characterization studies: a high-perfusion environment surrounding the implant, small size of pig brain, thick skull, uncertainty of positioning thermal mapping catheters and stationary probes deep into the brain, as well as limited time to keep the pigs alive under general anesthesia. Still, as observed in Figure 10, we were able to measure ever higher temperatures in the surrounding brain tissue with experience as we moved from Pig 1 to Pig 3. This was due to an improved surgical technique that allowed for the placement of thermal mapping catheters directly tangent to the balloon with no intervening tissue. The Pig 3 experiment demonstrated the ability to achieve our goal of ≥40 °C at 5 mm from the balloon surface, even with a small 1 cm diameter balloon. In human patients, we will be using larger balloons (3–5 cm) that will induce higher balloon temperatures. Thus, we expect to achieve therapeutic heating profiles up to 10 mm away to cover at-risk tissue around tumor resection cavities.

A major challenge for our radiation dose experiments involving 1 × 1 cm^2^ OSLD chips was their position uncertainty relative to the curved non-rigid balloon surface. Similarly, the position of temperature sensors inside the thermal mapping catheters and especially the free-floating stationary probe positions added a 1–2 mm uncertainty that explains the variability shown in the results for both the phantom and *in vivo* thermal experiments. Nonetheless, our data showed good reproducibility of the dosimetry across multi-trial experiments.

Specific therapeutic protocols for initial clinical trials of this approach have yet to be established. For GBM recurrences and brain metastases, we propose 21 Gy in 3 fractions based on the widely accepted hypo-fractionation dose of 35 Gy in 10 fractions for recurrent disease as per RTOG1205 [49]. For late toxicity of normal tissue with α/β ratio of 3 Gy [50], the biologically effective dose (BED) of brachytherapy is slightly lower than that of hypo-fractionation (70.0 vs. 75.8 Gy_3_, with the subscript “3” denoting α/β = 3 Gy). In regard to tumors, with α/β ratio of 8 Gy [51], the BED of brachytherapy is 22% lower than that of hypo-fractionation (39.4 vs. 50.3 Gy_8_). When accounting for the thermal enhancement of hyperthermia treatment, we expect that the BED of the TBT dose fractionation will be significantly higher, providing a notable therapeutic advantage as compared to hypo-fractionation radiation alone. While a recent study demonstrated the safety and efficacy of an intra-op brachytherapy platform in treating recurrent brain tumors and rapidly growing brain metastases [52], our dual-modality TBT approach could potentially achieve even better results.

In terms of newly diagnosed GBM, standard radiotherapy is EBRT with a 2 cm margin to account for microscopic disease and fractionation of 60 Gy in 30 fractions (5 fractions per week), starting 3–6 weeks after surgery in order for the skull to heal from the invasive craniotomy [53]. With this standard dose fractionation, the TBT balloon may be used as a local boost to regional EBRT. A recent study, however, revealed that about 95% of local GBM recurrences were within 1 cm distance from the initial T1-enhanced lesions [54], indicating that a 1 cm treatment margin might be adequate under certain circumstances. If this finding could be confirmed, it would be possible to treat certain GBM using TBT without a lengthy course of EBRT, and treatment would complete in one week immediately following surgery. 

## 5. Conclusions

Our laboratory studies in a static gel brain phantom, as well as our *in vivo* pig brain experiments, proved the feasibility and performance of prototype thermobrachytherapy (TBT) balloon implants for the simultaneous magnetic nanoparticle (MNP) hyperthermia and HDR brachytherapy of brain tumor resection cavities. Our data demonstrated the durability and robustness of TBT balloons, the compatibility of all heat and radiation delivery components, and the ability to generate well-localized heat and radiation dose distributions around the TBT balloons. Localizing treatment to the tumor margin should minimize detrimental effects in normal brain, while the capability of delivering *spherically symmetric* heat and radiation dose distributions *simultaneously* to the resection cavity margin should produce a maximal synergistic effect immediately after surgery and thereby improve clinical outcomes. Designed for intra-op applications, the TBT balloon device could accelerate therapy for brain cancers from months in and out of hospitals to one week following surgery, making it easier for patients and their families and more cost-effective for healthcare systems. Implementation of this device in human clinical trials could potentially lead to breakthrough therapy for resectable brain tumors. Following a successful launch in brain tumors, the TBT balloon procedure is expected to find application in other tumor resection sites including breast, head and neck, and sarcoma.

## 6. Patents

Thomas Jefferson University has patent pending technology with a published patent application for treating tumor bed margins of a resection cavity with simultaneous heat and radiation [55]. AMF Life Systems has patented technology for induction coils and systems for magnetic hyperthermia [42].

## Figures and Tables

**Figure 1 cancers-15-05683-f001:**
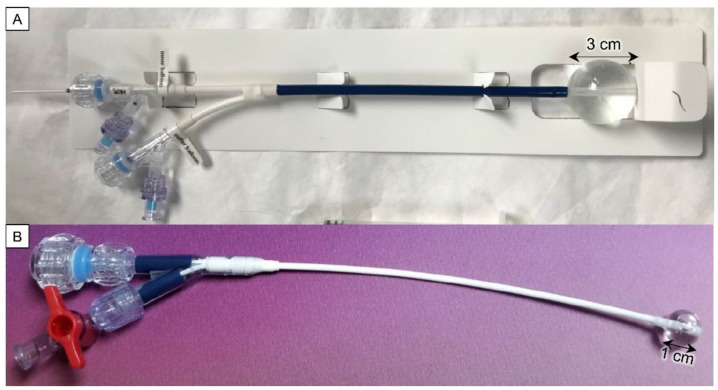
TBT implant prototypes filled with distilled water: (**A**) dual-wall balloon with 3 cm diameter and (**B**) single-wall balloon with 1 cm diameter. Note different colors of the flexible shafts (black for the 3 cm balloon and white for the 1 cm balloon).

**Figure 2 cancers-15-05683-f002:**
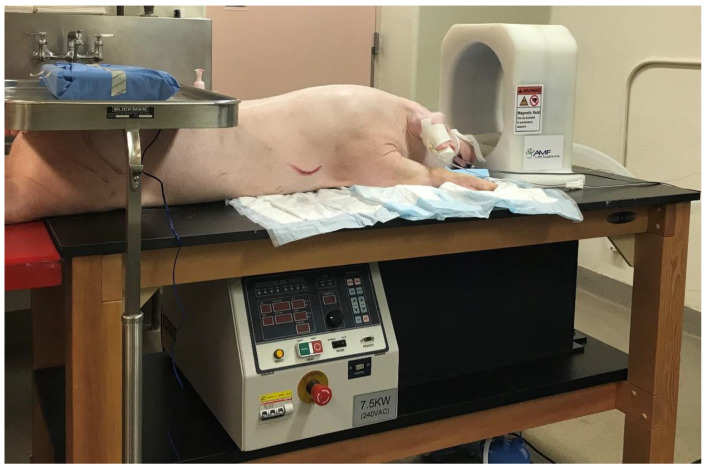
Experimental treatment table with a programmable 7.5 kW RF power generator underneath and a 133 kHz RF induction coil on top, and an anesthetized pig in preparation for surgical implantation of a TBT balloon for *in vivo* brain-heating experiment.

**Figure 3 cancers-15-05683-f003:**
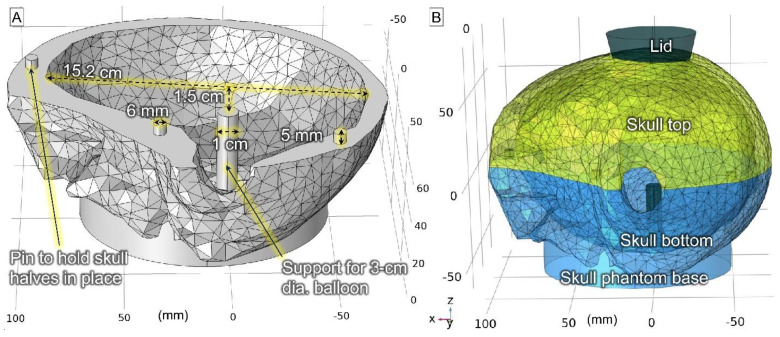
The 3D human skull model design. (**A**) bottom half showing the balloon support and 3 pins to secure the top half. (**B**) top and bottom halves assembled.

**Figure 4 cancers-15-05683-f004:**
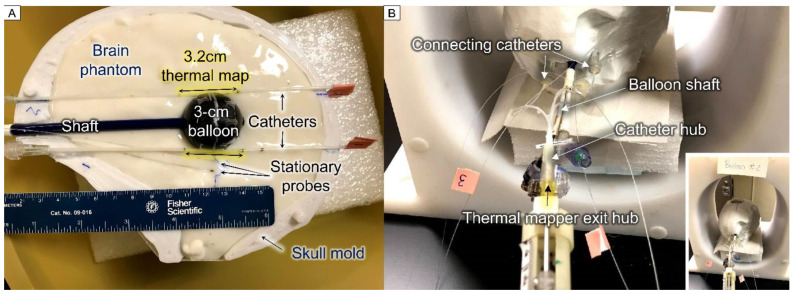
Setup photos for thermal dosimetry experiments: (**A**) bottom half of skull model filled with gel brain phantom displaying a 3 cm diameter TBT balloon filled with black EMG308 nanofluid, with 2 adjacent catheters for thermal mapping and 2 stationary probes positioned at about 5 and 10 mm distances from the balloon surface, and (**B**) fully assembled skull model inside the RF coil showing thermal mapping devices, stationary temperature probes, and balloon shaft; zoomed-out image in lower corner displays skull model centered in the RF coil.

**Figure 5 cancers-15-05683-f005:**
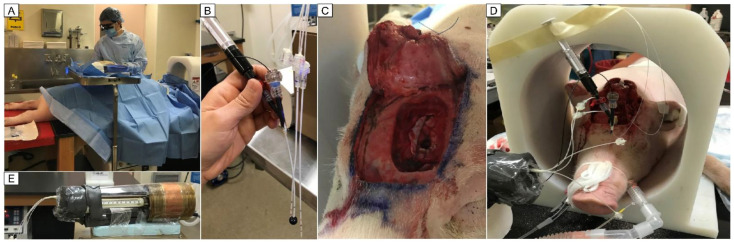
Experimental setup for the *in vivo* thermal dosimetry in a 40 kg anesthetized female pig: (**A**) pre-op setup with a neurosurgeon standing by; (**B**) a 1 cm diameter balloon specifically fabricated for pig brain, filled with black EMG304 nanofluid and attached to a 10 mL syringe for deflating and inflating the balloon, between two thermal mapping catheters; (**C**) corticectomy performed on the left hemisphere for insertion of the TBT balloon; (**D**) pig head in RF head coil with the balloon, thermal mapping catheters, and stationary probes inserted; and (**E**) thermal mapping motor with extra shielding added to minimize interference from the magnetic field.

**Figure 6 cancers-15-05683-f006:**
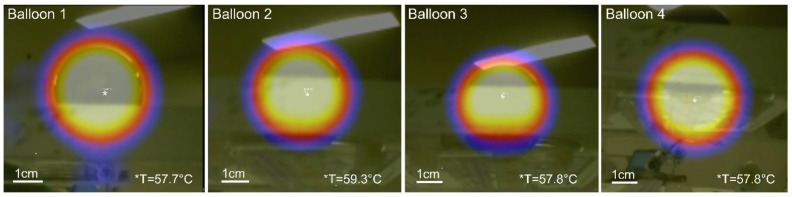
Temperature stress test of balloons in air: infrared images of four 3 cm diameter balloons with internal nanofluid temperature reaching 60 °C as recorded by fiber optic sensor. Temperature scale is not available for these images, only the maximum temperature was recorded by the infrared camera and displayed here. Background temperature was 22 °C.

**Figure 7 cancers-15-05683-f007:**
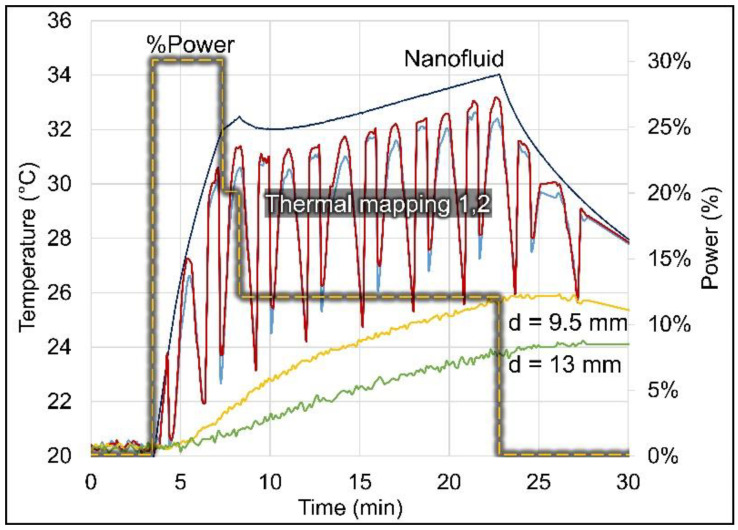
Temperature profiles measured in non-perfused brain phantom around a 3 cm diameter balloon (#3). The nanofluid temperature (dark blue line) was measured inside the balloon, the two thermal maps (red and blue lines) were measured inside catheters placed tangent to the balloon (radial distance to balloon surface ranged from 0.9–7.6 mm), and the stationary probes (yellow and green lines) were placed originally at 5 and 10 mm but moved to at 9.5 and 13 mm distances from the balloon surface. The relative RF power (shaded yellow dashed line) is shown on the secondary axis (Job #1, Table 2).

**Figure 8 cancers-15-05683-f008:**
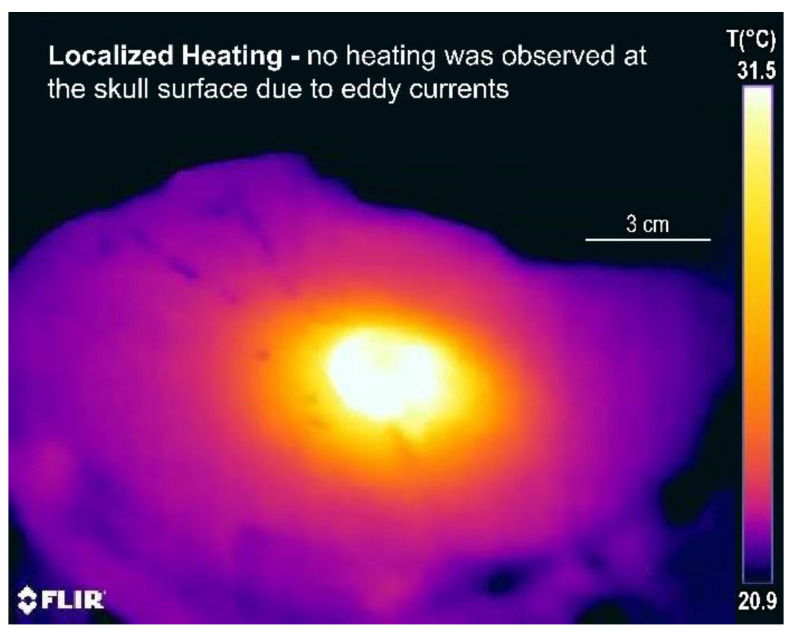
Split skull model heating test of a 3 cm diameter balloon: infrared thermal image of the bottom half of the skull model, 2 min after power off, showing well-localized heating around the balloon and no heating at the skull surface.

**Figure 9 cancers-15-05683-f009:**
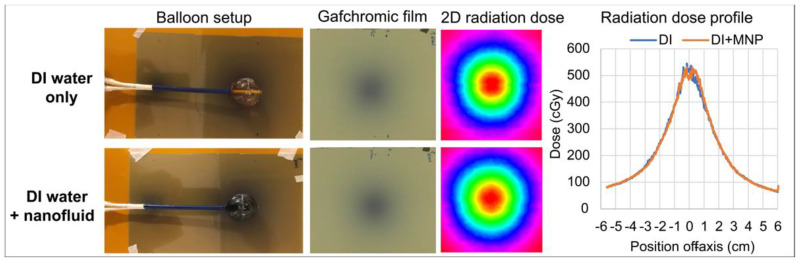
Gafchromic film measurements with a 3 cm balloon (#2). Top row: both chambers of the balloon filled with DI water (DI water only). Bottom row: inner chamber filled with DI water and outer chamber filled with nanofluid (DI water + nanofluid). Left: setup of balloon sitting on top of Gafchromic film. Center panels: 2D dose distribution captured on film with the corresponding color isodose plot. Right: comparison of dose profiles along the centerline of the film measurement plane.

**Figure 10 cancers-15-05683-f010:**
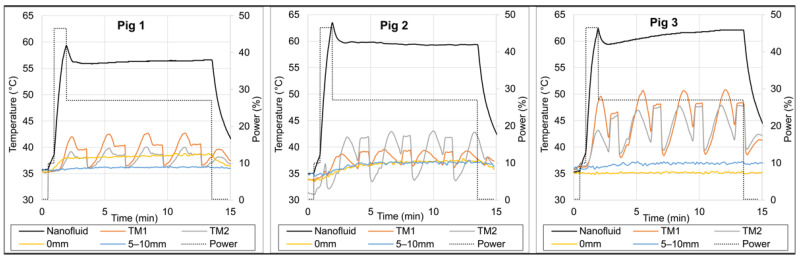
Temperature measurements in three live pig brain experiments. Nanofluid temperature (black line) was measured with a fiber optic sensor inside the 1 cm diameter balloon, thermal maps (TM1 orange and TM2 grey lines) were measured inside catheters tangent to the balloon (radial distance to balloon surface ranged from 0.9–4.9 mm), and the two stationary probes (yellow and blue lines) were placed at about 0 and 5–10 mm distances from the balloon surface. The relative power (grey dotted line) is shown on the secondary axis (Job #5, Table 2).

**Table 1 cancers-15-05683-t001:** Technical datasheet of magnetic nanoparticle solutions used in the present study.

Nanofluid	Saturation Magnetization (mT)	Viscosity at 27 °C (mPa·s)	Density at 25 °C (kg/m^3^)	Magnetite (Fe_3_O_4_) Particle Concentration (*v*/*v*)	Water Soluble Dispersant	Deionized Water
EMG 304	27.5	<10	1220	4.5%	1.5%	94.0%
EMG 308	6.6	<10	1050	1.2%	0.5%	98.3%

**Table 2 cancers-15-05683-t002:** Power sequences programmed to the RF power generator.

Job #	Relative Power P% (Duration)	H ^1^ (kA/m)	Application
1	30% (240 s), 20% (60 s), 12% (900 s)	5.6, 4.6, 3.5	Thermal mapping in phantom
2	15% (50 s), 35% (330 s), 10% (720 s)	4.0, 6.0, 3.2	In-air experiments
3	15% (50 s), 35% (200 s), 7% (720 s)	4.0, 6.0, 2.7	In-air experiments
4	15% (50 s), 55% (200 s), 20% (720 s)	4.0, 7.6, 4.6	In-phantom experiments
5	10% (30 s), 47% (60 s), 27% (720 s)	3.2, 7.0, 5.3	Thermal mapping in pig brain

^1^ Magnetic field strength at the center of the RF coil.

**Table 3 cancers-15-05683-t003:** Comparison of OSLD radiation dose readings on balloon#2 filled with DI water in both chambers (DI water) vs. the outer chamber filled with nanofluid (DI water + nanofluid).

Balloon #2 in Air		DI Water (cGy)	DI Water + Nanofluid (cGy)	Difference (%)
OSLD locations	distal	453	454	+0.2%
	above	532	451	−15.2%
	underneath	533	500	−6.2%
	left	538	453	−15.8%
	right	445	421	−5.4%
Average		500	456	−8.9%

**Table 4 cancers-15-05683-t004:** Comparison of OSLD radiation dose readings on the surface of balloons #1–4 in air, without magnetic field exposure (HDR) and with magnetic field turned on to activate nanofluid heating (HDR + HT).

In air		7 January 2021			8 January 2021		
		HDR (cGy)	HDR + HT (cGy)	Difference (%)	HDR (cGy)	HDR + HT (cGy)	Difference (%)
Balloon #2		D_A_ = 30 mm × D_L_ = 34 mm, T_max_ = 77 °C			D_A_ = 29 mm × D_L_ = 34 mm, T_max_ = 55 °C		
OSLD locations	Distal	438	380	−13.2%	430	408	−5.1%
	Top	455	480	+5.5%	440	469	+6.6%
	Bottom	515	481	−6.6%	495	498	+0.6%
	Left	481	467	−2.9%	482	435	−9.8%
	Right	380	384	1.1%	397	379	−4.5%
**Average**		**454**	**438**	**−3.4%**	**449**	**438**	**−2.5%**
OSLD underneath balloon		HDR (cGy)	HDR + HT (cGy)	Difference (%)	HDR (cGy)	HDR + HT (cGy)	Difference (%)
Balloon #1		D_A_ = 29 mm × D_L_ = 34 mm, T_max_ = 66 °C			D_A_ = 30.0 mm × D_L_ = 33.6 mm, T_max_ = 52.2 °C		
Dose (cGy)		607	536	−11.7%	527	449	−14.8%
Balloon #3		D_A_ = 30.8 mm × D_L_ = 29.4 mm, T_max_ = 64.5 °C			D_A_ = 28.0 mm × D_L_ = 30.3 mm, T_max_ = N/A		
Dose (cGy)		512	487	−4.9%	N/A due to leaking from injection site		
Balloon #4		D_A_ = 28.5 mm × D_L_ = 33 mm, T_max_ = 64.2 °C			D_A_ = 28.5 mm × D_L_ = 33.5 mm, T_max_ = 53.4 °C		
Dose (cGy)		498	459	−7.8%	430	481	+11.9%

**Table 5 cancers-15-05683-t005:** Comparison of OSLD radiation dose readings on the surface of balloon #3 in air, without magnetic field exposure (HDR) and with magnetic field turned on to activate nanofluid heating (HDR + HT). OSLDs were placed directly underneath the balloon and on top of a 1 cm bolus, i.e., at radial distance r = 0 and 10 mm from the balloon surface, respectively.

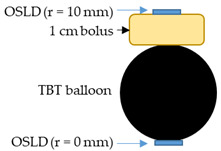	Balloon #3 in air, D_A_ = 30.7 mm × D_L_ = 30.5 mm, T_max_ = 52 °C			
	Radial distance from the balloon surface	HDR (cGy)	HDR + HT (cGy)	Difference (%)
	r = 0 mm (directly underneath the balloon)	506	521	+3.0%
	r = 10 mm (converted from 0 mm distance)	186	191	+3.0%
	r = 10 mm (on top of 1 cm bolus)	182 *	179	−1.6%

* When performing OSLD measurements with HDR alone, the tape that fixed the OSLD to the 1 cm bolus released on one side and the OSLD detached about 1.5 mm from the 1 cm bolus. Thus, we corrected the dose for the additional 1.5 mm separation, converting the reading of 151 cGy obtained at 11.5 mm to 182 cGy at 10.0 mm radial distance from the balloon surface.

**Table 6 cancers-15-05683-t006:** Comparison of OSLD radiation dose readings on the surface of balloons #1–4 in phantom, without magnetic field exposure (HDR) and with magnetic field turned on to activate nanofluid heating (HDR + HT).

In Phantom	Balloon #1			Balloon #2			Balloon #3			Balloon #4		
Dimensions	D_A_ = 30.2 mm D_L_ = 33.5 mm			D_A_ = 29.3 mm D_L_ = 33.6 mm			D_A_ = 30.7 mm D_L_ = 30.5 mm			D_A_ = 30.5 mm D_L_ = 33.5 mm		
T_max_	43 °C			49 °C			48 °C			49.5 °C		
OSLD locations	HDR (cGy)	HDR + HT (cGy)	Difference (%)	HDR (cGy)	HDR + HT (cGy)	Difference (%)	HDR (cGy)	HDR + HT (cGy)	Difference (%)	HDR (cGy)	HDR + HT (cGy)	Difference (%)
Top	537	535	−0.4%	496	480	−3.2%	514	420	−18.3%	401	424	+5.7%
Bottom	496	506	+2.0%	557	513	−7.9%	462	433	−6.3%	517	462	−10.6%
Left	401	516	+28.7%	563	497	−11.7%	487	477	−2.1%	441	425	−3.6%
Right	429	410	−4.4%	466	458	−1.7%	466	457	−1.9%	437	458	+4.8%
**Average**	**466**	**492**	**+5.6%**	**520**	**487**	**−6.4%**	**482**	**447**	**−7.4%**	**449**	**442**	**−1.5%**

## Data Availability

Data are contained within the article.

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
