# Peer review of "Evaluation of a Balloon Implant for Simultaneous Magnetic Nanoparticle Hyperthermia and High-Dose-Rate Brachytherapy of Brain Tumor Resection Cavities"

_cancers, 2023, doi:10.3390/cancers15235683_

Round 1
Reviewer 1 Report
Comments and Suggestions for Authors
In the Introduction, authors provide a brief review of hyperthermia (HT) combinations with radiotherapy and/or brachytherapy for treatment of brain malignancies. This review of the state of art provides no mention of standard of care. Multiple references describing trials using interstitial HT (by various technical modalities) are cited, with little distinction by (HT) modality. The statement that trials were conducted with equipment that was never "deemed ready for widespread dissemination" is unclear, considering at least one of the interstitial HT modalities cited, magnetic hyperthermia, was approved (in 2010 by European Medicines Agency) for widespread use in treating recurrent glioblastoma with EBRT.
That current standard of care for treating glioblastoma (GBM) and other brain malignancies does not include magnetic hyperthermia (or any other HT modality) is not mentioned. Readers should be provided a frank and clear assessment of the state of art with more nuanced motivation for the presented innovation in light of the historical attempts and outcomes.
In the Results, the reported field amplitude was identified by an algorithm preferenceing a max H value, however, no measured data or calibration were provided. Measured magnetic field parameters (e.g. frequency and amplitude) with variance should be reported.
Discussion lacks critical analysis of results with an assessment of the study limitations, and prospects for clinical translation.
Comments on the Quality of English LanguageOverall, English is acceptable but will benefit from minor edits. For example:
Page 7, line 239 - "...gently pressuring..." ought to instead be "...gently pressing..."
Page 15, Line 514 (and other places) - use of "regiment" is inappropriate.
Author Response
Dear Reviewer,
Thank you for your excellent feedback. Please see our response in the attached PDF file.
Kind regards,
Authors

Reviewer 2 Report
Comments and Suggestions for Authors
The study presented here is a significant advancement in the field of glioblastoma treatment and thermal therapy. Here are the reasons for my positive evaluation:
- Innovative Approach: The study introduces a novel thermobrachytherapy (TBT) balloon implant that combines magnetic nanoparticle (MNP) hyperthermia with high‐dose‐rate (HDR) brachytherapy, which is an innovative and promising approach for the treatment of glioblastoma.
- Robustness and Compatibility: The authors have thoroughly evaluated the robustness of the balloon device and the compatibility of its heat and radiation delivery components. This comprehensive assessment instills confidence in the feasibility of this treatment modality.
- Data Quality: The quality of the data presented in this manuscript is impressive. The 3D‐printed skull model for in‐phantom heating and radiation measurements, as well as the in vivo experiments in pig brains, provide a strong foundation for the potential clinical applications of this technology.
- Thermal Mapping: The demonstration of spherically symmetric heating in both phantom and brain tissue, along with the ability to heat well‐perfused brain tissue to hyperthermic levels (> 40°C), showcases the effectiveness of the TBT balloon implant.
- Clinical Promise: The conclusion that the device is now ready for finalization of regulatory approvals in anticipation of early‐stage clinical investigation is a significant step forward in potentially improving clinical outcomes for patients with glioblastoma.
Author Response

(The authors gave the same response as above.)

Reviewer 3 Report
Comments and Suggestions for Authors
The manuscript titled “Evaluation of Balloon Implant for Simultaneous Magnetic Nanoparticle Hyperthermia and High-Dose-Rate Brachytherapy of Brain Tumor Resection Cavities” by Wan, S.; et al. is an original scientific work where the authors study the implementation of a thermobrachytherapy balloon in model pigs. This ballon enables de delivery of magnetic nanoparticles and thus, the synergistic effect between hyperthermia and high-dose-rate brachytherapies. For it, the authors quantitatively tested (in vivo and in vitro experiments) the thermal response to the balloon heating by external magnetic fields. The work is highly innovative and it is expected to be patented to pursue the clinical Trials.
However, it exists some points that need to be addressed (please, see them below detailed point-by-point). The most relevant outcomes found by the authors can contribute in the growth of many fields like the linked to the healthcare, overall focused in the treatment of human glioblastoma tumoral disorders. For this reason, I will recommend the present scientific manuscript for further publication in Cancers once all the below described suggestions will be properly fixed.
Here, there exists some points that must be covered in order to improve the scientific quality of the manuscript paper:
1) KEYWORDS (OPTIONAL). The authors should consider to add the term “pig model” in the keyword list.
2) INTRODUCTION. “While each of these technologies (…) technical challenges (…) the heat optimally with radiation” (lines 72-75). Here,the authors outline the potential limitations related to the injection of magnetic nanoparticles. The authors should not discharge the risk of body accumulation which could lead hepatotoxic effects [1] and some recent reported approaches to overcome this tissular toxicity [2].
[1] Soares, G.A.; Pereira, G.M.; Romualdo, G.R.; Biasotti, G.G.A.; Stoppa, E.G.; Bakuzis, A.F.; Baffa, O.; Barbisan, L.F.; Miranda, J.R.A. Biodistribution Profile in Magnetic Nanoparticles in Cirrhosis-Associated Hepatocarcinogenis in Rats by AC Biosusceptometry. Pharmaceutics 2022, 14, 1907. https://doi.org/10.3390/pharmaceutics14091907.
[2] Cerqueira, M.; Belmonte-Reche, E.; Gallo, J.; Baltazar, F.; Bañobre-López, M. Magnetic Solid Nanoparticles and Their Counterparts: Recent Advances towards Cancer Theranostics. Pharmaceutics 2022, 14, 506. https://doi.org/10.3390/pharmaceutics14030506.
3) MATERIALS & METHODS. “In this study, we used two magnetic nanoparticle (…) (1.2% v/v; density of 1050 kg/m3).” (lines 149-150). What is the purity degree of the used MNPs? Did the authors test this property by chemical analysis? In case affirmative, this information should be provided at least as Supplementary Information (SI). Otherwise, this information could be available in the respective datasheets from the supplier.
4) Table 1 (line 156). The authors found two different saturation magnetization values (27.5 mT and 6.6 mT, respectively). Since this parameter directly relies on the geometry and size of the MNPs, did the authors characterize their dimensions (e.g. transmission electron microscopy measurements) and morphologies? A brief statement should be provided in this regard.
5) “To mimic brain tissue (…) 6.545% TX-151 powder (…) and 0.455% NaCel (w/w).” (lines 212-213). Please, the authors should homogenize the significant figures with respect to the shown in the Table 1. This point should be considered for the rest of the main manuscript body text.
6) “This design was then 3D-printed using polylactic acid filament” (lines 218-219). The authors prototyped a pig skull with this material. Polylactic acid (PLA) treasures numerous advantages for this purpose but also some limitations (e.g. low flexural strength). Could this negatively affect to the further studies conducted in this prototype material? The authors should furnish a brief statement in this regard.
7) “The procedure to load the balloon (…) the 3 cm TBT balloon (…) thermometry setup” (lines 221-223) and Figure 4 (line 250). Did the authors quantitatively test the increase of intracraneal pressure caused by the introduction of the TBT balloon inside the pig skull? This value is in agreement with other craneal surgeries? A brief statement should be provided in this regard (during the upcoming 2.3. subsection in lines 312-343).
8) “Experiment 1 evaluates the effect of surrounding nanofluid on radiation dose (…) effects of magnetic field and heating of the balloon (…) radiation dose” (lines 291-293). Here, the authors show the strategy used to measure the impact of external magnetic fields on the balloon thermal performance. A brief discussion with other alternative techniques like magneto-optical Kerr effect (MOKE) [3] or single molecule approaches [4] should be furnished to highlight the advantages of the approach selected by the authors (e.g. possibility to devote longitudinal studies to track the patient progress over time, reproducibility to gather feasible results, …).
[3] Yamamoto, S.; Matsuda, I. Measurement of the Resonant Magneto-Optical Kerr Effect Using a Free Electron Laser. Appl. Sci. 2017, 7, 662. https://doi.org/10.3390/app7070662.
[4] Winkler, R.; Ciria, M.; Ahmad, M.; Plank, H.; Marcuello, C. A Review of the Current State of Magnetic Force Microscopy to Unravel the Magnetic Properties of Nanomaterials Applied in Biological Systems and Future Direction for Quantum Technologies. Nanomaterials 2023, 13, 2585. https://doi.org/10.3390/nano13182585.
9) RESULTS. Figure 6 (line 358). The authors should add a lateral scale bar in this Figure. Same comment for the Figure 8 (line 387).
10) Table 4 (line 430). What was the number of measurements carried out at certain condition (population size) taken into account by the authors? The standard deviation (SD) values should be detailed in this Table. Same comment for the Table 5 (line 433) and Table 6 (line 442).
11) Thermal Dosimetry in Pig Brain (lines 446-460). Nanofluid temperature of Pigs 1 and 2 exhibits a plateau after reaching the maximum peak. This observation does not happen for the third individual pig where a small increase of temperature arrived after this peak. The authors should add a brief discussion about this point.
12) DISCUSSION. This section perfectly remarks the most significant outcomes found in this work. No actions are requested for the authors.
13) CONCLUSIONS. “Implementation of this device in human clinical trials could potentially lead to breakthrough therapy for resectable brain tumors” (lines 552-553). Could this technology be extandable for other types of cancer or malignancies? A brief statement should be provided in this regard.
14) REFERENCES. The references are not in the proper format style of Cancers. The authors should take care of this point.
Author Response

(The authors gave the same response as above.)
